# FedARC: Adaptive Residual Compensation for Data and Model Heterogeneous Federated Learning

## Abstract

Federated learning (FL) enables multiple clients to collaboratively train models without sharing private data, but practical FL is hindered by both data heterogeneity and model heterogeneity. Existing Heterogeneous FL (HtFL) methods often suffer from inadequate representation alignment and limited knowledge transfer, especially under fine-grained distribution shifts, thus limiting both personalization and generalization. To remedy this, we propose FedARC, a novel HtFL framework with Adaptive Residual Compensation. FedARC adaptively fuses local and global representations through a trainable projector and applies dynamic residual correction to mitigate feature-level distribution mismatches. Moreover, FedARC incorporates semantic anchor alignment to further reduce inter-client feature divergence, thereby stabilizing knowledge transfer and aggregation. We theoretically prove FedARC converges with a non-convex convergence rate $\mathcal{O}(1/T)$. Extensive experiments on five public benchmarks demonstrate that FedARC surpasses nine state-of-the-art HtFL baselines by up to 2.63% in average accuracy, while maintaining efficient communication and computation.

## 1 Introduction

Federated learning (FL) enables multiple clients to train collaboratively without sharing raw data. In practice, however, classical algorithms such as FedAvg (McMahan et al., 2016) assume a single architecture across clients, while real deployments exhibit mixed data distributions, device capabilities, and model designs. Specifically, client data are often non-IID (Lu et al., 2024), devices range from phones to edge servers with disparate compute/communication budgets (Yi et al., 2022), and organizations maintain proprietary architectures to meet internal or IP constraints (Shao et al., 2023). These factors jointly make a "one-size-fits-all" global model brittle (Qi et al., 2023).

Rather than enforcing a single global architecture, heterogeneous federated learning (HtFL) (Tan et al., 2021; Zhang et al., 2025) embraces architectural diversity and non-IID data, enabling collaboration via transferable knowledge, thus is widely regarded as a more realistic and increasingly mainstream paradigm in FL. Existing HtFL approaches can be broadly grouped into three categories: (i) *Knowledge-distillation (KD)–based* methods transfer logits/representations using public or proxy data or summary statistics (Morafah et al., 2024; Park et al., 2023; Wu et al., 2021; Zhu et al., 2021); they reduce coupling to a shared architecture but often suffer from limited or biased proxies, added communication or server-side computation, and potential privacy risks. (ii) *Model-split* methods share only a homogeneous part (e.g., a header or an extractor) while keeping heterogeneous parts local (Yi et al., 2023); fixed partitioning can constrain expressiveness and may leak architectural priors (Yang et al., 2024). (iii) *Mutual-learning* methods co-train a large heterogeneous model together with a small homogeneous proxy per client and aggregate only the proxy (Shen et al., 2020; Weng et al., 2025); yet capacity mismatch and insufficient calibration leave knowledge transfer suboptimal and increase compute and communication costs (Yi et al., 2024a). However, a shared failure mode persists across these categories: batch-level distribution shifts and fine-grained feature misalignment are often under-addressed, causing misaligned class centers and overlapping clusters that hurt both generalization and personalization. As illustrated in Fig. 1(a,c), features from conventional HtFL scatter and drift across clients, inducing domain bias and unstable aggregation.

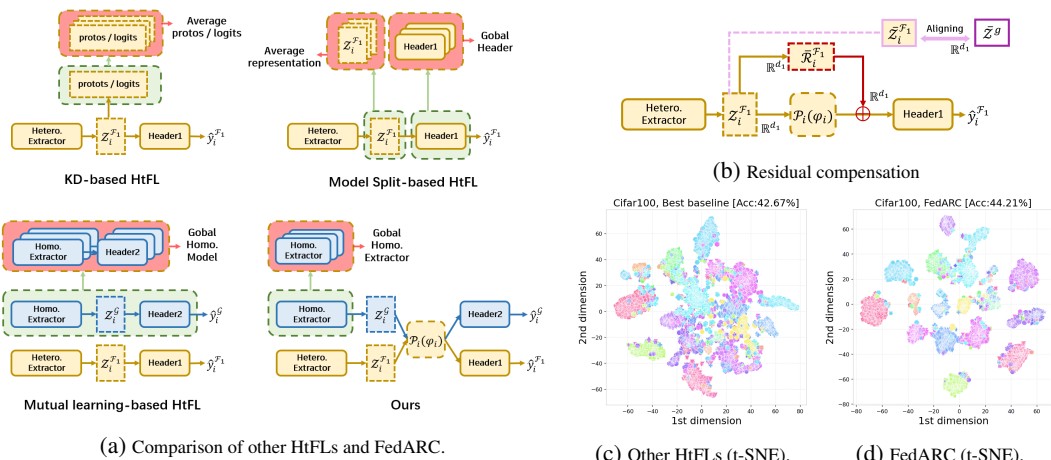

(a) Comparison of other HtFLs and FedARC.

(b) Residual compensation

(c) Other HtFLs (t-SNE).

(d) FedARC (t-SNE).

Figure 1: KD-based HtFL: clients share prediction statistics (logits/prototypes); the server averages them and broadcasts guidance for the next round. Model-split HtFL: a unified component (e.g., shared header) is aggregated across clients while the rest stays local. Mutual-learning HtFL: a small homogeneous model is co-trained with clients and aggregated to exchange global knowledge. Ours: only the small homogeneous extractor is uploaded/aggregated; on-client a lightweight projector fuses features and residual compensation (see (b)) calibrates the fused representation before either head consumes it, reducing cross-client feature drift. (Red: server-aggregated; green: clients-upload items; yellow: local heterogeneous model/representationsmodule; blue: global homogeneous model/representationsmodule.)

To address these limitations, we propose FedARC—Federated learning with Adaptive Residual Compensation—an HtFL framework that fuses local and global representations via a trainable projector and performs dimension-wise residual compensation to correct distribution shifts within and across clients. A semantic-anchor alignment module further reduces inter-client divergence and stabilizes aggregation. As illustrated in Fig. 1,(c), conventional HtFL misaligns local semantic centers (client means), leading to scattered or overlapping features and domain bias that hampers cross-client transfer. In contrast, FedARC (Fig. 1,(d)) explicitly compensates local representation shifts with residual vectors and anchors each client's overall feature mean to a global semantic anchor, yielding compact, well-separated clusters in a shared semantic space while maintaining global consistency. Consequently, FedARC achieves a balanced trade-off between personalization and generalization by preserving local semantics while aligning to global anchors. The fusion mechanism, combined with residual compensation, adaptively bridges the heterogeneity gap, whereas semantic-anchor alignment provides a stable cross-client target.

Our main contributions are summarized as follows:

- We introduce a residual compensation for fine-grained fusion and bidirectional knowledge transfer, enabling effective batch-level personalization in the presence of data and model heterogeneity. By aligning representations with global semantic anchors, FedARC reduces inter-client feature divergence and stabilizes aggregation, further enhancing generalization.

- We provide a theoretical analysis showing that FedARC achieves a non-convex convergence rate of $\mathcal{O}(1/T)$ under standard smoothness and bounded-variance assumptions.

- Extensive experiments on five benchmark datasets under both statistical and model heterogeneity settings demonstrate that FedARC outperforms nine state-of-the-art HtFL methods by up to 2.63% in average accuracy, while maintaining communication and computation efficiency.

## 2 RELATED WORK

Existing HtFL methods can be broadly classified as (i) partially heterogeneous, where clients extract submodels from a unified template via pruning or reconfiguration (e.g., EMO (Wu et al., 2025), FCCL+ (Huang et al., 2023), InCo (Chan et al., 2023), FedSA-LoRA (Guo et al., 2024), DepthFL (Kim et al., 2023)); or (ii) fully heterogeneous, where each client uses a distinct architecture, requiring specialized knowledge fusion strategies, which can be further divided as follows:

**Knowledge distillation (KD)-based HtFL.** Existing KD strategies for HtFL fall into three categories: (i) *Statistical transfer*: Methods such as FD (Jeong et al., 2018), FedProto (Tan et al., 2021), PLADA (Fang et al., 2024), FedTGP (Zhang et al., 2024b), FedHCD (Feng et al., 2024), and FedSA (Zhou et al., 2025) exchange only summary statistics or prototypes, reducing bandwidth but still risking membership inference. Beyond, FedLabel (Cho et al., 2023) selectively assimilates local or global predictions for semi-supervised FL but still assumes homogeneous models and works at the logit/pseudo-label level. (ii) *Proxy logit synchronization*: Systems such as FedGD (Zhang et al., 2023), FZSL (Sun et al., 2024), DFRD (Luo et al., 2023), and FedDTG (Gong et al., 2023) generate synthetic anchors for alignment, but face extra optimization, mode collapse, and privacy leakage if synthetic data memorize local features. (iii) *Synthetic data distillation*: Approaches like Fed-ET (Cho et al., 2022), FSFL (Huang et al., 2022), KRR-KD (Park et al., 2023), and TAKFL (Morafah et al., 2024) synchronize soft outputs using a proxy dataset, but this incurs high communication and exposes models to privacy attacks (e.g., PLI (Takahashi et al., 2023)).

**Model Split-based HtFL.** These methods split each local model into feature extractor and task-specific header. In FedRep (Collins et al., 2021), FedPAC (Xu et al., 2023), FedAS (Yang et al., 2024), and FedAlt/FedSim (Pillutla et al., 2022), aggregate homogeneous *headers* while keeping heterogeneous extractors. Others, such as LG-FedAvg (Liang et al., 2020), FedGen (Zhu et al., 2021), FedGH (Yi et al., 2023), and CHFL (Liu et al., 2022), aggregate homogeneous *extractors* for consistent representation while keeping headers private, while $CD^2$-pFed (Shen et al., 2022) focusing on parameter partitioning rather than cross-architecture alignment. However, partial model sharing may inadvertently reveal architectural priors and constrain task adaptability.

**Mutual learning-based HtFL.** These methods co-train each client's full-size heterogeneous model with a lightweight homogeneous auxiliary; the auxiliary enables server-side aggregation while the large model preserves local inductive bias (Xu et al., 2024; Zhang et al., 2024a; Louizos et al., 2024). Representative systems (pFedES (Yi et al., 2025), FedKD (Wu et al., 2021), FedMRL (Yi et al., 2024b)), Fed-CO2 (Cai et al., 2023) and Fed-RoD (Chen & Chao, 2022) verify this idea but roughly double compute/communication. Variants (FedAPEN (Qin et al., 2023), FedSKD (Weng et al., 2025), pFedAFM (Yi et al., 2024a)) improve aggregation flexibility yet still transfer low-dimensional or indirect signals, limiting fusion fidelity.

Distinctly, our FedARC framework keeps both data and local heterogeneous models on client, aggregating only a lightweight homogeneous extractor at the server, enabling robust global fusion while preserving personalization. Crucially, FedARC bridges heterogeneity via fine-grained residual compensation with batch-wise anchor alignment, dynamically aligning global and local representations.

## 3 PRELIMINARIES

FedAvg (McMahan et al., 2016) is the canonical FL algorithm, where a central server coordinates $N$ clients. At the beginning of each communication round, the server randomly selects a fraction $\rho$ of $N$ clients, yielding an active set $S$ with cardinality $K = \rho \cdot N$. The server broadcasts the current global model, denoted as $\mathcal{F}(\omega)$ ($\mathcal{F}(\cdot)$ is model structure and $\omega$ its parameters). Each selected client $k$ trains the received model on its local dataset $\mathcal{D}_k \sim P_k$ ($\mathcal{D}_k$ obeys distribution $P_k$, datasets across clients are generally non-IID). Using gradient descent, the parameters are updated as $\omega_k \leftarrow \omega - \eta \nabla \ell \left( \mathcal{F}\left( \boldsymbol{x}_i; \omega \right), y_i \right)$, where $\ell(\cdot, \cdot)$ is the sample-wise loss for $(\boldsymbol{x}_i, y_i) \in \mathcal{D}_k$. After local training, each client uploads the updated model parameters $\omega_k$ to the server. The server refines the global model by weighted aggregation, $\omega = \sum_{k \in S} \frac{n_k}{n} \omega_k$ ($n_k = |\mathcal{D}_k|$ is the number of data samples on client $k$, $n = \sum_{k=0}^{N-1} n_k$ is the number of total data samples on all clients).

FedAvg assumes identical architectures and minimizes the average loss of the global model $\mathcal{F}(\omega)$,

$$\min_{\omega \in \mathbb{R}^d} \sum_{k=0}^{N-1} \frac{n_k}{n} \mathcal{L}_k \Big( \mathcal{F}(\omega); \mathcal{D}_k \Big), \tag{1}$$

where $d$ is the parameter dimension of $\omega$, and $\mathcal{L}_k$ is the average loss of global model $\mathcal{F}(\omega)$ on $\mathcal{D}_k$.

In this work, all clients tackle the same prediction task, yet each maintains its own distinct local model with different architecture, $\mathcal{F}_k(\cdot)$ with personalized model parameters $\omega_k$. Our FedARC

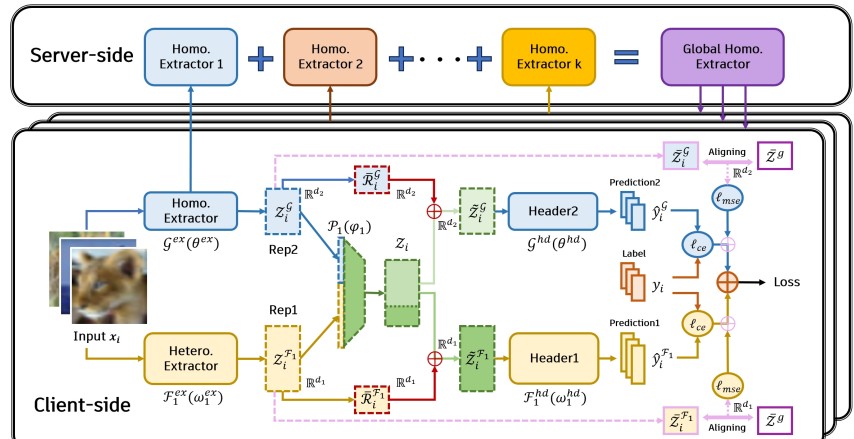

Figure 2: Workflow of FedARC. For each client $k$, the heterogeneous extractor $\mathcal{F}_k^{ex}(\omega_k^{ex})$ and the shared homogeneous extractor $\mathcal{G}^{ex}(\theta^{ex})$ encode the same input $x_i$ into local and global features $\mathcal{Z}_i^{\mathcal{F}_k}$ and $\mathcal{Z}_i^{\mathcal{G}}$. A lightweight projector $\mathcal{P}_k(\varphi_k)$ fuses these features into $\mathcal{Z}_i$, which is further adjusted by residual vectors $\bar{\mathcal{R}}_i^{\mathcal{F}_k}$ and $\bar{\mathcal{R}}_i^{\mathcal{G}}$ and anchored to the global mean $\bar{\mathcal{Z}}^g$ to correct client-specific distribution shifts. The compensated representations are consumed by the heterogeneous head $\mathcal{F}_k^{hd}$ and homogeneous head $\mathcal{G}^{hd}$ to compute local objectives, while only the parameters of $\mathcal{G}^{ex}$ (blue) are uploaded and aggregated on the server; and local heterogeneous model (yellow) remain on-device.

framework aims to minimize the cumulative loss of these personalized models on their local data:

$$\min_{\omega_k \in \mathbb{R}^{d_k}} \sum_{k=0}^{N-1} \mathcal{L}_k\left(\mathcal{F}_k\left(\omega_k\right); \mathcal{D}_k\right), \tag{2}$$

where the dimensionality $d_k$ vary across all clients.

## 4 THE PROPOSED FEDARC APPROACH

To enable effective knowledge transfer under HtFL, we assign a lightweight global homogeneous model $\mathcal{G}(\theta)$ ($\mathcal{G}(\cdot)$ is the structure, $\theta$ denotes parameters) to all clients.

**Homogeneous Feature Extractor Structure.** Assigning a homogeneous feature extractor ensures computational tractability on clients and supports global knowledge consolidation. As intermediate features capture richer semantics than output logits, we use a compact CNN-based extractor as the global shared module. Only the extractor's parameters are exchanged and aggregated at the server, enabling effective knowledge fusion while keeping other model components local and private.

### 4.1 OVERVIEW

Each communication round $t$ in FedARC is as follows:

1. The server randomly samples $K$ clients from $N$ participants and broadcasts the latest global homogeneous feature extractor $\mathcal{G}^{ex}(\theta^{ex,t-1})$ to each selected client in $\mathcal{S}^t$.

2. Each participating client $k$ jointly trains its local heterogeneous extractor $\mathcal{F}_k^{ex}(\omega_k^{ex,t-1})$ and the received global homogeneous extractor $\mathcal{G}^{ex}(\theta^{ex,t-1})$ on local dataset $(x_i, y_i)$ from $\mathcal{D}_k$. For each batch, both extractors generate their respective feature representations. These representations are then separately fed into the local header $\mathcal{F}_k^{hd}(\omega_k^{hd,t-1})$ for the heterogeneous branch and the global header $\mathcal{G}^{hd}(\theta^{hd,t-1})$ for the homogeneous branch.

3. Once local training concludes, only the updated homogeneous feature extractor $\mathcal{G}^{ex}(\theta_k^{ex,t})$ is uploaded to the server. The server performs a weighted aggregation to update the global homogeneous feature extractor $\mathcal{G}^{ex}(\theta^{ex,t})$.

These steps repeat until all clients' heterogeneous local models $\mathcal{F}_k(\omega_k)$ converge. During inference, only each client's personalized local heterogeneous model is used. Further details are provided in Algorithm 1 (Appendix).

Building on Eq.(2), the training objective is reformulated as minimizing the sum of the loss of the combined model $\mathcal{C}_k(\varepsilon_k) = \mathcal{F}_k(\omega_k) \circ \mathcal{G}(\theta)$ across all clients:

$$\min_{\omega_0,\ldots,\omega_{N-1},\theta} \sum_{k=0}^{N-1} \mathcal{L}_k\Big(\mathcal{C}_k(\omega_k \circ \theta); \mathcal{D}_k\Big). \tag{3}$$

## 4.2 Feature Representation Fusion

**Motivation.** A straightforward approach is to feed global and local feature representations into their respective headers and jointly optimize them by minimizing the sum of their losses. However, this suffers from two issues: (1) Low-quality representations from the global extractor might exacerbate misalignment with local features; (2) The absence of feature calibration allows local representation bias to accumulate over time, hindering effective knowledge transfer and generalization. To address this, FedARC employs adaptive residual compensation and semantic anchor alignment for fine-grained alignment and fusion of local and global knowledge at the feature level.

## 4.3 Adaptive Residual Compensation

Given a training sample $(\boldsymbol{x}_i, y_i) \in \mathcal{D}_k$, $\boldsymbol{x}_i$ is processed by the heterogeneous extractor $\mathcal{F}_k^{ex}(\omega_k^{ex,t-1})$ and the global homogeneous extractor $\mathcal{G}^{ex}(\theta_k^{ex,t-1})$ to obtain two respective representations,

$$\mathcal{Z}_i^{\mathcal{F}_k} = \mathcal{F}_k^{ex}(\boldsymbol{x}_i; \omega_k^{ex,t-1}), \quad \mathcal{Z}_i^{\mathcal{G}} = \mathcal{G}^{ex}(\boldsymbol{x}_i; \theta^{ex,t-1}), \tag{4}$$

where $\mathcal{Z}_i^{\mathcal{F}_k} \in \mathbb{R}^{d_1}$ contains personalized local knowledge specific to the client's data distribution, $\mathcal{Z}_i^{\mathcal{G}} \in \mathbb{R}^{d_2}$ contains generalized knowledge beneficial across all clients. Inspired by widely used knowledge fusion methods (Yi et al., 2024b;a), we concatenate local and global feature representations and project them via a learnable projector $\mathcal{P}_k(\varphi_k^{t-1})$:

$$\mathcal{Z}_i = \mathcal{P}_k\big(\mathcal{Z}_i^{\mathcal{F}_k} \circ \mathcal{Z}_i^{\mathcal{G}}; \varphi_k^{t-1}\big), \quad \mathcal{P}_k : \mathbb{R}^{d_1+d_2} \to \mathbb{R}^{d_1} \tag{5}$$

Due to significant heterogeneity in both data and model architectures, direct feature concatenation may still leave latent semantic discrepancies. To explicitly mitigate this, we consider *client-level* adaptive residual vectors, $\bar{\mathcal{R}}_i^{\mathcal{F}_k} \in \mathbb{R}^{d_1}$ and $\bar{\mathcal{R}}_i^{\mathcal{G}} \in \mathbb{R}^{d_2}$, learned during training to adjust each client's representation space:

$$\widetilde{\mathcal{Z}}_i^{\mathcal{F}_k} = \mathcal{Z}_i^{1:d_1} + \bar{\mathcal{R}}_i^{\mathcal{F}_k}, \quad \widetilde{\mathcal{Z}}_i^{\mathcal{G}} = \mathcal{Z}_i^{1:d_2} + \bar{\mathcal{R}}_i^{\mathcal{G}}. \tag{6}$$

The local heterogeneous headers takes the full fused $\mathcal{Z}_i^{1:d_1} \in \mathbb{R}^{d_1}$, while the homogeneous headers operates on its prefix subspace $\mathcal{Z}_i^{1:d_2} \in \mathbb{R}^{d_2}$ with $d_1 > d_2$. This *nested slicing* preserves a shared subspace for global alignment without disjointly splitting features. These residual compensations are jointly optimized with model parameters, enabling each client to flexibly align fused representations with their unique local semantic spaces $\mathbb{R}^{d_1}, \mathbb{R}^{d_2}$. Each client's heterogeneous prediction header $\mathcal{F}_k^{hd}(\omega_k^{hd})$ and the global homogeneous prediction header $\mathcal{G}^{hd}(\theta^{hd})$ output logits in $\mathbb{R}^L$ ($L$ is the label dimension):

$$\hat{y}_i^{\mathcal{F}_k} = \mathcal{F}_k^{hd}(\widetilde{\mathcal{Z}}_i^{\mathcal{F}_k}; \omega_k^{hd,t-1}), \quad \hat{y}_i^{\mathcal{G}} = \mathcal{G}^{hd}(\widetilde{\mathcal{Z}}_i^{\mathcal{G}}; \theta^{hd,t-1}). \tag{7}$$

The prediction losses (cross-entropy (Zhang et al., 2018)) are computed separately for each branch:

$$\ell_i^{\mathcal{F}_k} = \ell_{ce}(\hat{y}_i^{\mathcal{F}_k}, y_i), \quad \ell_i^{\mathcal{G}} = \ell_{ce}(\hat{y}_i^{\mathcal{G}}, y_i). \tag{8}$$

## 4.4 Semantic Anchor Alignment

To mitigate semantic drift and further separate $\mathcal{Z}_i$ and $\bar{\mathcal{R}}_i$, we propose a semantic anchor alignment that encourages the learnable projector to produce $\mathcal{Z}_i$ by a more client-invariant mean, while the residual vectors $\bar{\mathcal{R}}_i^{\mathcal{F}_k}$ remain client-specific offsets that are not directly constrained by this anchor.

Concretely, we regularize the means $\bar{\mathcal{Z}}_i^{\mathcal{F}_k} \in \mathbb{R}^{d_1}$ and $\bar{\mathcal{Z}}_i^{\mathcal{G}} \in \mathbb{R}^{d_2}$ by aligning them with a global consensus mean $\bar{\mathcal{Z}}^g$ for each feature dimension independently. The global mean $\bar{\mathcal{Z}}^g = \left(\sum_{k=1}^{N} n_k\right)^{-1} \sum_{k=1}^{N} n_k \bar{\mathcal{Z}}_k^g, \forall \bar{\mathcal{Z}}^g \in \mathbb{R}^{d_1}$, is determined during the FL initialization phase. We measure the similarity between each client's mean ($\bar{\mathcal{Z}}_i^{\mathcal{F}_k}$, $\bar{\mathcal{Z}}_i^{\mathcal{G}}$) and the global anchor ($\bar{\mathcal{Z}}^{g,1:d_1} \in \mathbb{R}^{d_1}$, $\bar{\mathcal{Z}}^{g,1:d_2} \in \mathbb{R}^{d_2}$) by the mean squared error (MSE) (Tüchler et al., 2002), with hyperparameter $\lambda$ controlling regularization strength. Empirically, this yields the following objective, updating Eq. (8):

$$\ell_i^{\mathcal{F}_k} = \ell_{ce}(\hat{y}_i^{\mathcal{F}_k}, y_i) + \lambda \cdot \ell_{mse}(\bar{\mathcal{Z}}_i^{\mathcal{F}_k}, \bar{\mathcal{Z}}^{g,1:d_1}), \quad \ell_i^{\mathcal{G}} = \ell_{ce}(\hat{y}_i^{\mathcal{G}}, y_i) + \lambda \cdot \ell_{mse}(\bar{\mathcal{Z}}_i^{\mathcal{G}}, \bar{\mathcal{Z}}^{g,1:d_2}). \quad (9)$$

In the above, the semantic anchor terms require empirical mean estimates. For each client, we calculate $\bar{\mathcal{Z}}_i^{\mathcal{F}_k}$ : $\hat{\bar{\mathcal{Z}}}_i^{\mathcal{F}_k} = \frac{1}{n_k} \sum_{j=1}^{n_k} \mathcal{F}_k^{ex}(\boldsymbol{x}_{ij}; \omega_k^{ex})$ and $\bar{\mathcal{Z}}_i^{\mathcal{G}}$ : $\hat{\bar{\mathcal{Z}}}_i^{\mathcal{G}} = \frac{1}{n_k} \sum_{j=1}^{n_k} \mathcal{G}^{ex}(\boldsymbol{x}_{ij}; \theta^{ex})$ over the entire local dataset. However, SGD only accesses mini-batches in each forward pass, we utilize a moving average strategy to approximate these statistics, following (Li et al., 2019; Zhang et al., 2014) we obtain:

$$\hat{\bar{\mathcal{Z}}}_i^{\mathcal{F}_k} = (1 - \kappa) \cdot \hat{\bar{\mathcal{Z}}}_i^{\mathcal{F}_k, t-1} + \kappa \cdot \hat{\bar{\mathcal{Z}}}_i^{\mathcal{F}_k, t}, \quad \hat{\bar{\mathcal{Z}}}_i^{\mathcal{G}} = (1 - \kappa) \cdot \hat{\bar{\mathcal{Z}}}_i^{\mathcal{G}, t-1} + \kappa \cdot \hat{\bar{\mathcal{Z}}}_i^{\mathcal{G}, t}, \quad (10)$$

where $\hat{\bar{\mathcal{Z}}}_i^{\mathcal{F}_k, t-1}$, $\hat{\bar{\mathcal{Z}}}_i^{\mathcal{G}, t-1}$ are from the previous batch, and $\hat{\bar{\mathcal{Z}}}_i^{\mathcal{F}_k, t}$, $\hat{\bar{\mathcal{Z}}}_i^{\mathcal{G}, t}$ are from the current batch, $\kappa$ serves as a momentum coefficient that balances historical and current batch statistics. Since the two feature extractors are updated locally but may be reset between global rounds, these averages are recomputed at the start of each communication round and not carried across rounds.

Finally, we compute the total loss by weighting the two branches using coefficients $\alpha_i^{\mathcal{F}_k}$ and $\alpha_i^{\mathcal{G}}$:

$$\ell_i = \alpha_i^{\mathcal{F}_k} \cdot \ell_i^{\mathcal{F}_k} + \alpha_i^{\mathcal{G}} \cdot \ell_i^{\mathcal{G}}. \quad (11)$$

By default, both weights are set to $\alpha_i^{\mathcal{F}_k} = \alpha_i^{\mathcal{G}} = 1$ to ensure equal contribution. The composite loss $\ell_i$ is jointly optimized for the heterogeneous local model, global feature extractor, and projector using SGD following FedAvg,

$$\omega_k^t \leftarrow \omega_k^{t-1} - \eta_\omega \nabla \ell_i, \quad \theta_k^t \leftarrow \theta^{t-1} - \eta_\theta \nabla \ell_i, \quad \varphi_k^t \leftarrow \varphi_k^{t-1} - \eta_\varphi \nabla \ell_i, \quad (12)$$

where the learning rates $\eta_\omega, \eta_\theta, \eta_\varphi$ are set identically to ensure convergence stability. This unified training procedure encourages representations to be simultaneously generalizable and personalized. In implementation, we update $\bar{\mathcal{R}}^{\mathcal{F}_k}$, $\bar{\mathcal{R}}^{\mathcal{G}}$ with the same SGD (omitted in Eq. (12) for brevity).

## 5 CONVERGENCE ANALYSIS

We denote notations following (Yi et al., 2025; 2024b). Let $t$ denote the communication round, and $e \in \{0, 1, ..., E\}$ the iterations of local training within each round. At the start of round $t + 1$ (iteration $tE + 0$), client $k$ receives the global homogeneous feature extractor $\mathcal{G}^{ex}(\theta^{ex,t})$ from the server. The $e$-th local update is indexed by $tE + e$, and $tE + E$ indicates the completion of local training in round $t + 1$, after which the client uploads its updated $\mathcal{G}^{ex}(\theta_k^{ex,t+1})$ for aggregation. We modify client $k$'s combined local model as $\mathcal{C}_k(\varepsilon_k) = (\mathcal{F}_k(\omega_k) \circ \mathcal{G}(\theta) | \mathcal{P}_k(\varphi_k))$, where $\mathcal{F}_k(\omega_k)$ is the local heterogeneous model, $\mathcal{G}(\theta)$ is the global small homogeneous model, and $\mathcal{P}_k(\varphi_k)$ is the learnable projector. The learning rate for each component is $\eta = \{\eta_\omega, \eta_\theta, \eta_\varphi\}$.

**Assumption 1.** *Lipschitz Smoothness. The gradients of client $k$'s combined local heterogeneous model $\varepsilon_k$ are $L1$–Lipschitz smooth (Tan et al., 2021),*

$$\left\| \nabla \mathcal{L}_k^{t_1}\left(\varepsilon_k^{t_1}; \boldsymbol{x}, y\right) - \nabla \mathcal{L}_k^{t_2}\left(\varepsilon_k^{t_2}; \boldsymbol{x}, y\right) \right\| \leqslant L_1 \left\| \varepsilon_k^{t_1} - \varepsilon_k^{t_2} \right\|, \quad (13)$$
$$\forall t_1, t_2 > 0, k \in \{0, 1, \ldots, N-1\}, (\boldsymbol{x}, y) \in \mathcal{D}_k.$$

*The above formulation can be re-expressed as:*

$$\mathcal{L}_k^{t_1} - \mathcal{L}_k^{t_2} \leqslant \left\langle \nabla \mathcal{L}_k^{t_2}, \left(\varepsilon_k^{t_1} - \varepsilon_k^{t_2}\right) \right\rangle + \frac{L_1}{2} \left\| \varepsilon_k^{t_1} - \varepsilon_k^{t_2} \right\|_2^2. \quad (14)$$

**Assumption 2.** *Unbiased Gradient and Bounded Variance. For each client $k$, the stochastic gradient $g_{\varepsilon,k}^t = \nabla \mathcal{L}_k^t(\varepsilon_k^t; \mathcal{B}_k^t)$ computed over mini-batch $\mathcal{B}_k^t$ is unbiased:*

$$\mathbb{E}_{\mathcal{B}_k^t \subseteq \mathcal{D}_k}\left[g_{\varepsilon,k}^t\right] = \nabla \mathcal{L}_k^t\left(\varepsilon_k^t\right), \quad (15)$$

*and the variance of $g_{\varepsilon,k}^t$ is bounded by:*

$$\mathbb{E}_{\mathcal{B}_k^t \subseteq \mathcal{D}_k}\left[\left\|\nabla \mathcal{L}_k^t\left(\varepsilon_k^t; \mathcal{B}_k^t\right) - \nabla \mathcal{L}_k^t\left(\varepsilon_k^t\right)\right\|_2^2\right] \leqslant \sigma^2. \quad (16)$$

**Assumption 3.** *Bounded Parameter Variation. The parameter variations of the small homogeneous feature extractor $\theta_k^t$ and $\theta$ before and after aggregation at the FL server is bounded by,*

$$\left\| \theta^t - \theta_k^t \right\|_2^2 \leq \delta^2. \tag{17}$$

Given the above assumptions, we obtain the following results (proofs provided in the Appendix).

**Lemma 1.** *Under Assumptions 1 and 2, for arbitrary client, the loss during local iterations $\{0, 1, ..., E\}$ in the $t + 1$-th training round is bounded by:*

$$\mathbb{E}\left[\mathcal{L}_{(t+1)E}\right] \leq \mathcal{L}_{tE+0} + \left(\frac{L_1\eta^2}{2} - \eta\right) \sum_{e=0}^{E} \|\nabla\mathcal{L}_{tE+e}\|_2^2 + \frac{L_1 E\eta^2\sigma^2}{2}. \tag{18}$$

**Lemma 2.** *Under Assumptions 2 and 3 after $(t + 1)$-th local training round, the loss of any client before and after aggregating the small homogeneous feature extractors at FL server is bounded by:*

$$\mathbb{E}\left[\mathcal{L}_{(t+1)E+0}\right] \leq \mathbb{E}\left[\mathcal{L}_{tE+1}\right] + \eta\delta^2. \tag{19}$$

**Theorem 1.** *One Complete Round of FL. Based on the Lemma 1 and Lemma 2, for any client, after local training, model aggregation and receiving the new global homogeneous feature extractor, we have:*

$$\mathbb{E}\left[\mathcal{L}_{(t+1)E+0}\right] \leq \mathcal{L}_{tE+0} + (\frac{L_1\eta^2}{2} - \eta) \sum_{e=0}^{E} \|\nabla\mathcal{L}_{tE+e}\|_2^2 + \frac{L_1 E\eta^2\sigma^2}{2} + \eta\delta^2. \tag{20}$$

**Theorem 2.** *Non-convex Convergence Rate of FedARC. Based on the above assumptions and lemma, for any client and an arbitrary constant $\epsilon > 0$:*

$$\frac{1}{T} \sum_{t=0}^{T-1} \sum_{e=0}^{E-1} \|\nabla\mathcal{L}_{tE+e}\|_2^2 \leq \frac{\frac{1}{T}\sum_{t=0}^{T-1}\left(\mathcal{L}_{tE+0} - \mathbb{E}(\mathcal{L}_{(t+1)E+0})\right)}{\eta - \frac{L_1\eta^2}{2}} + \frac{\frac{L_1 E\eta^2\sigma^2}{2} + \eta\delta^2}{\eta - \frac{L_1\eta^2}{2}} < \epsilon,$$

$$s.t. \ \eta < \frac{2\left(\epsilon - \delta^2\right)}{L_1\left(\epsilon + E\sigma^2\right)} \tag{21}$$

Consequently, under the conditions outlined above, for any client's local model in FedARC is guaranteed to achieve a non-convex convergence rate of $\epsilon \sim \mathcal{O}(1/T)$, provided that the learning rates for the local heterogeneous model, homogeneous feature extractor, and learnable projector satisfy the specified constraints.

# 6 EXPERIMENTAL EVALUATION

**Datasets.** We evaluate our approach on five widely used image-classification datasets, including Cifar10/100 (Krizhevsky, 2009), Flowers102 (Nilsback et al., 2008), Tiny-ImageNet (Chrabaszcz et al., 2017), and DomainNet (Peng et al., 2018).

**Statistical/Model heterogeneity.** To simulate statistical heterogeneity, we partition client data using a Dirichlet distribution as in prior works (Li et al., 2021b;a; Zhang et al., 2025). Specifically, for each class $c$ and client $k$, $q_{c,k} \sim \text{Dir}(\beta)$ (with $\beta = 0.1$ by default) determines the proportion of samples from class $c$ assigned to client $k$. Each client's local data is split 75% for training and 25% for testing, and evaluation is performed on the test set. To simulate model heterogeneity among clients, we follow FedTGP (Zhang et al., 2024b), where each client is equipped with a heterogeneous model architecture. This setting is denoted as "HtFE$^{img}{}_X$", with $X$ indicating the number of different feature extractor architectures in the FL. Each client $k$ is assigned the $(k \bmod X)$-th architecture from the pool. Specifically, for the "HtFE$^{img}{}_8$" setting in our main experiments, we employ eight diverse architectures: 4-layer CNN (McMahan et al., 2016), GoogLeNet (Szegedy et al., 2014), MobileNetV2 (Sandler et al., 2018), and ResNet18/34/50/101/152 (He et al., 2015). To ensure feature dimension consistency across architectures, we append an average pooling layer after each feature extractor, resulting in a unified output dimension $d_1$ (set to 512 by default, $d_2$ set to 256). The homogeneous model used for all clients is a 4-layer CNN.

**Implementation details.** We benchmark FedARC against 9 best baseline methods, each representing a prominent approach from the three main branches of fully model-heterogeneous FL, as detailed

Table 1: Average test accuracy (%) on four datasets in cross-silo and cross-device settings under label skew using $\text{HtFE}^{img}_8$.

| Settings | Cross-silo (N=10, $\rho$=100%, $\beta$=0.1) | | | | Cross-device (N=50, $\rho$=20%, $\beta$=0.1) | | | |
|---|---|---|---|---|---|---|---|---|
| Datasets | Cifar10 | Cifar100 | Flowers102 | Tiny* | Cifar10 | Cifar100 | Flowers102 | Tiny* |
| FD | 87.78±0.14 | 42.67±0.07 | 53.31±0.65 | 25.95±0.07 | 84.07±0.15 | 39.62±0.09 | 47.82±0.54 | 24.57±0.12 |
| FedProto | 84.04±0.19 | 36.12±0.10 | 40.12±0.18 | 19.31±0.12 | 67.89±0.41 | 20.82±0.07 | 35.08±0.19 | 18.73±0.21 |
| FedTGP | 87.64±0.12 | 40.69±0.19 | 55.57±0.23 | 27.43±0.08 | 80.57±0.24 | 37.34±0.10 | 47.57±0.26 | 25.64±0.17 |
| LG-FedAvg | 86.47±0.10 | 40.16±0.07 | 46.39±0.39 | 26.08±0.06 | 83.07±0.28 | 38.01±0.08 | 41.21±0.42 | 24.56±0.05 |
| FedGen | 85.34±0.23 | 38.68±0.15 | 45.05±0.16 | 20.68±0.08 | 80.89±0.33 | 36.18±0.12 | 41.27±0.21 | 18.86±0.13 |
| FedGH | 86.07±0.16 | 41.61±0.08 | 47.31±0.15 | 25.41±0.12 | 82.91±0.23 | 37.63±0.09 | 42.85±0.21 | 25.18±0.15 |
| pFedES | 87.54±0.28 | 42.37±0.26 | 51.42±0.26 | 26.87±0.31 | 83.64±0.32 | 38.77±0.35 | 47.77±0.46 | 25.33±0.33 |
| FedKD | 87.64±0.14 | 41.91±0.21 | 50.48±0.25 | 26.26±0.16 | 83.16±0.17 | 38.36±0.11 | 44.86±0.23 | 24.93±0.19 |
| FedMRL | 87.20±0.26 | 42.57±0.32 | 51.65±0.21 | 27.37±0.28 | 83.78±0.31 | 38.93±0.33 | 45.65±0.24 | 25.43±0.30 |
| FedARC | **89.22±0.07** | **44.21±0.05** | **57.67±0.11** | **29.91±0.14** | **86.14±0.14** | **42.25±0.07** | **50.18±0.15** | **27.10±0.16** |

Table 2: The test accuracy (%) on Cifar100 under the cross-device setting with various model heterogeneity. $\Delta$: The largest accuracy difference among $\text{HtFE}^{img}_2$, $\text{HtFE}^{img}_3$, $\text{HtFE}^{img}_5$ and $\text{HtFE}^{img}_9$

| Settings | Heterogeneous Feature Extractors | | | | | Heterogeneous Models | | |
|---|---|---|---|---|---|---|---|---|
| | $\text{HtFE}^{img}_2$ | $\text{HtFE}^{img}_3$ | $\text{HtFE}^{img}_5$ | $\text{HtFE}^{img}_9$ | $\Delta$ | $\text{Res34-HtC}^{img}_4$ | $\text{HtFE}^{img}_8\text{-HtC}^{img}_4$ | $\text{HtM}^{img}_{10}$ |
| FD | 42.88±0.15 | 40.92±0.36 | 39.98±0.13 | 36.62±0.31 | 6.26 | 41.27±0.15 | 38.76±0.08 | 39.82±0.05 |
| FedProto | 35.63±0.22 | 29.89±0.48 | 27.75±0.64 | 21.91±0.28 | 13.72 | 28.62±0.21 | 21.75±0.71 | 32.26±0.13 |
| FedTGP | 42.14±0.29 | 37.97±0.57 | 37.66±0.42 | 36.84±0.61 | **5.30** | 42.21±0.36 | 39.35±0.17 | 38.71±0.18 |
| LG-FedAvg | 41.22±0.13 | 40.59±0.42 | 39.66±0.23 | 35.39±0.11 | 5.83 | - | - | - |
| FedGen | 40.05±0.54 | 38.47±0.29 | 38.48±0.28 | 34.67±0.12 | 5.38 | - | - | - |
| FedGH | 40.11±0.18 | 39.82±0.21 | 38.14±0.18 | 34.38±0.31 | 5.73 | - | - | - |
| pFedES | 42.16±0.12 | 40.69±0.57 | 38.93±0.31 | 36.47±0.48 | 5.69 | 41.07±0.38 | 38.96±0.27 | **40.04±0.25** |
| FedKD | 42.19±0.09 | 40.53±0.07 | 37.47±0.21 | 35.82±0.25 | 6.37 | 37.57±0.45 | 37.89±0.48 | 37.63±0.13 |
| FedMRL | 42.27±0.09 | 40.76±0.61 | 39.42±0.24 | 36.55±0.42 | 5.72 | 41.21±0.42 | 39.58±0.25 | 39.73±0.21 |
| FedARC | **44.36±0.12** | **42.72±0.08** | **41.28±0.06** | **37.28±0.18** | 7.08 | **42.91±0.09** | **40.39±0.13** | 39.74±0.10 |

in the Related Work section. Knowledge distillation: FD (Jeong et al., 2018), FedProto (Tan et al., 2021) and FedTGP (Zhang et al., 2024b). Model split: LG-FedAvg (Liang et al., 2020), FedGen (Zhu et al., 2021) and FedGH (Yi et al., 2023). Mutual learning: pFedES (Yi et al., 2025), FedKD (Wu et al., 2021) and FedMRL (Yi et al., 2024b). We consider two widely used FL settings, the cross-silo and the cross-device setting (Kairouz et al., 2019). In the cross-silo setting, we employ a federation of 10 clients, all of which participate in each communication round (i.e., participation ratio $\rho = 1$). In the cross-device setting, we increase the number of clients to 50, with a random subset of 20% (i.e., $\rho = 0.2$) selected in each round. The cross-device setting serves as the primary focus of our study, as it closely mirrors practical federated deployments involving heterogeneous, resource-constrained edge devices. Unless explicitly specified, we follow previous work (Zhang et al., 2024b), each selected client performs a single local epoch of training per round, utilizing a mini-batch size of 10 and a unified learning rate of $\eta_\omega = \eta_\theta = \eta_\varphi = 0.01$. All experiments run for 1,000 communication rounds and are repeated three times.

## 6.1 Comparison Results

**Average Accuracy.** As shown in Table 1, FedARC achieves the best average accuracy across all benchmarks, outperforming 9 state-of-the-art HtFL baselines by up to 2.63% in both cross-silo and cross-device settings. While the gap with the competitive baseline (FD) on Tiny*(Tiny-ImageNet) is small, Appendix Figure 1 visualizes the accuracy curves of FedARC and FD on Cifar10/100, showing that FedARC not only converges to higher final accuracy, but also matches or exceeds FD in convergence speed. By aligning auxiliary and local embeddings in a shared subspace and regularizing class means, FedARC corrects feature misalignment that logit-only distillation (e.g., FD, FedKD) overlooks under data and model heterogeneity. Compared with prototype methods (e.g., FedProto, FedTGP), its anchor-based first-order alignment avoids enforcing a single global prototype, adapts to client drift, and improves accuracy and convergence. All further experiments focus on the more challenging Cifar10/100 datasets to rigorously evaluate HtFL performance.

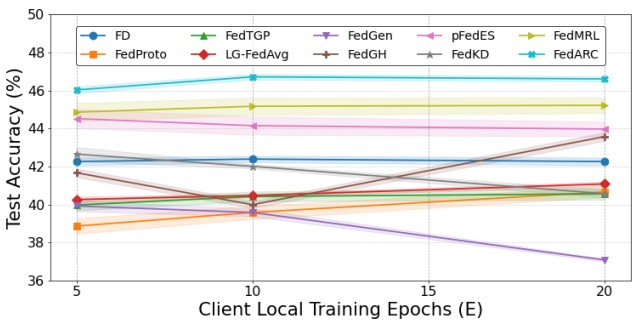

Table 3: Ablation studies on the key components of FedARC on Cifar100 in the HtFE$^{img}_8$ setting. FRF: feature representation fusion. ARC: adaptive residual compensation. SAA: semantic anchor alignment.

| FRF | ARC | SAA | Acc(%) |
|:---:|:---:|:---:|:---:|
| ✓ | | | 40.13 |
| ✓ | ✓ | | 43.64 (↑ **3.51**) |
| ✓ | | ✓ | 41.39 (↑ **1.26**) |
| ✓ | ✓ | ✓ | **44.21** (↑ **4.08**) |

Figure 3: Test accuracy (%) on Cifar100 in the cross-silo setting using HtFE$^{img}_8$ with large local training epochs $E$.

Table 4: Test accuracy (%) on Cifar100 in HtFE$^{img}_8$ with different $\lambda$ and $\kappa$ (default: $\lambda = 1$, $\kappa = 0.1$).

| | | $\kappa = 0.1$ | | | | $\lambda = 1$ | | |
|:---:|:---:|:---:|:---:|:---:|:---:|:---:|:---:|:---:|
| | $\lambda = 0.1$ | $\lambda = 1$ | $\lambda = 5$ | $\lambda = 10$ | $\kappa = 0.05$ | $\kappa = 0.1$ | $\kappa = 0.5$ | $\kappa = 1$ |
| Acc. | 42.92±0.13 | **44.21±0.05** | 43.14±0.37 | 42.46±0.81 | 43.65±0.10 | **44.21±0.05** | 41.87±0.08 | 40.46±0.15 |

**Impact of Model Heterogeneity.** As shown in Table 2, FedARC achieves the best performance in the cross-device settings under both statistical and model heterogeneity. Compared to FD, which distills from averaged global logits without addressing feature misalignment, FedARC leverages projection-based fusion and residual compensation to better align semantics across clients. FedTGP improves separability via prototypes, but its fixed anchors adapt poorly as heterogeneity increases. FedGH is efficient but the backbone–head decoupling hurts generalization. The best mutual learning baselines (e.g., FedMRL, pFedES) align predictions but not structure, risking representation collapse. In contrast, FedARC explicitly aligns heterogeneous representations, enabling more stable and transferable learning under both data and model heterogeneity.

**Impact of Local Training Epochs.** Increasing the number of client-side local epochs ($E$) can reduce the number of communication rounds in FL. As shown in Figure 3, mutual learning methods (e.g., pFedES and FedKD) degrade as $E$ grows. We attribute this to their reliance on an auxiliary model: longer local training amplifies client-specific bias in the auxiliary branch, which then propagates during aggregation. In contrast, FedMRL and FedARC mitigate this by fusing features from the auxiliary and local models. Moreover, FedARC leverages residual compensation and anchor alignment to regularize client features, yielding the highest accuracy across all $E$.

**Ablation and Hyperparameter Analysis.** Table 3 shows ablation results on the key modules of FedARC. ARC on the top of FRF brings a significant accuracy gain (+3.51%), while SAA further improves performance (+1.26%). Combining all components achieves the best result (44.21%), highlighting complementary roles of ARC and SAA. Table 4 analyzes hyperparameter sensitivity under cross-silo setting. The optimal accuracy is reached with $\lambda = 1$ and $\kappa = 0.1$; either too large or too small values degrade performance, demonstrating that FedARC is robust across broad hyperparameter range and consistently outperforms other HtFL methods even in less optimal settings.

**Communication and Computation Costs.** We measure communication overhead as the total upload and download bytes per round using the float32 data type in PyTorch, and computation as the average GPU time for each client and server on idle GPUs. From Table 5, we observe: (1) Mutual learning methods, despite transmitting smaller models, still incur high communication, and SVD in FedKD does not bring notable savings. (2) KD-based methods are communication-efficient but limited by the lower information capacity of prototypes/logits, resulting in lower accuracy. (3) FedGen and FedTGP involve extra server-side training and multiple rounds, leading to higher computational power consumption on the server than other HtFLs. Overall, FedARC offers the best trade-off—highest accuracy with no higher communication than other mutual learning methods.

**Impact of Feature Dimensions.** Figure 4 shows that accuracy generally improves as the feature dimension $d_1$ increases from 64 to 256. Partial parameter–sharing methods (LG-FedAvg, FedGen) deviate from this trend. Most methods peak at $d_1 = 256$, whereas FedMRL and FedGH continue

Table 5: Communication and computation costs on Cifar100 using HtFE$^{img}_8$. MB (megabytes), s (seconds).

| Items | Comm. (MB) | | Computation (s) | |
|---|---|---|---|---|
| | Up. | Down. | Client | Server |
| FD | 0.52 | 0.89 | 6.54 | 0.04 |
| FedProto | 3.17 | 5.01 | 6.68 | 0.05 |
| FedTGP | 3.17 | 5.01 | 6.61 | 7.91 |
| LG-FedAvg | 5.81 | 5.81 | 6.22 | 0.05 |
| FedGen | 5.81 | 30.22 | 5.79 | 3.02 |
| FedGH | 3.17 | 4.81 | 9.75 | 0.41 |
| pFedES | 39.87 | 39.87 | 17.63 | 0.07 |
| FedKD | 43.24 | 43.24 | 8.14 | 0.08 |
| FedMRL | 50.75 | 50.75 | 8.27 | 0.08 |
| FedARC | 39.87 | 39.87 | 7.83 | 0.08 |

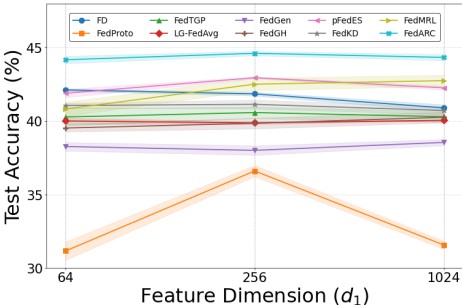

Figure 4: Test accuracy (%) on Cifar100 in the Dirichlet setting using HtFE$^{img}_8$ with varying feature dimensions $d_1$.

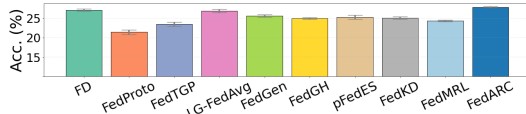

Figure 5: Test accuracy (%) on DomainNet under the feature shift scenario using HtFE$^{img}_5$.

to rise at $d_1 = 1024$, while others (e.g., FD, FedProto) plateau or decline, suggesting that excessively large feature spaces can complicate optimization or invite overfitting—especially for standard aggregation or KD-based methods. Notably, FedARC remains best across all $d_1$.

**Performance in the Feature Shift Setting.** From Figure 5, FedARC delivers the superior results, surpassing FD and LG-FedAvg. Prototype-sharing methods degrade notably under cross-domain features, while mutual learning baselines cluster in a similar mid-range. FedProto shows a significant performance gap, as large cross-domain style gaps, feature means diverge, making the global prototypes biased and prone to drift without explicit alignment.

## 7 CONCLUSION

This paper proposed a novel HtFL approach, FedARC, based on sharing homogeneous feature extractors with efficient privacy preservation, and communication and computational cost savings. It enables each client to alternatively train a homogeneous feature extractor and heterogeneous local model to exchange global and local knowledge. Aggregating the homogeneous local feature extractors from clients fuses knowledge across heterogeneous clients. Theoretical analysis and experiments demonstrate its effectiveness and efficiency in both communication and computation.

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
