# OpenReview forum: "FedARC: Adaptive Residual Compensation for Data and Model Heterogeneous Federated Learning"
_ICLR.cc/2026/Conference — Submitted to ICLR 2026_

### Official Review · Reviewer_74tW · 2025-10-15

**Soundness:** 2
**Presentation:** 2
**Contribution:** 3
**Rating:** 4
**Confidence:** 4

**Summary:**

FedARC is a Heterogeneous Federated Learning (HtFL) framework that enhances both personalization and generalization by adaptively compensating for feature mismatches between clients. It fuses local and global representations through a trainable projector, applies dynamic residual correction, and aligns semantic anchors to stabilize knowledge transfer. Experiments on four benchmarks show consistent improvements over state-of-the-art baselines.

**Strengths:**

- Empirical results across various benchmarks demonstrate that the proposed method outperforms baselines.
- Model heterogeneity in federated learning is a practical scenario that reflects real-world heterogeneity.

**Weaknesses:**

- Lack of explanation in the overview figure (Fig. 2): Since Fig. 2 lacks sufficient explanation in the caption, it is difficult to understand the overall workflow of the proposed method.

- Unrealistic data heterogeneity: In real-world scenarios, each client may specialize in distinct benchmarks even when performing the same task (e.g., classification). For example, client 1 may focus on ImageNet-1K, while client 2 focuses on CIFAR-10. Although splitting client data using a Dirichlet distribution is a common approach, evaluating the proposed method under more realistic data heterogeneity would better demonstrate its robustness.

- Limited model diversity: While the paper includes various architectures (e.g., ResNet18/34/50/101/152 and MobileNet), these are all relatively simple convolutional models. To validate the generalizability of the proposed method, it would be beneficial to include experiments with heterogeneous ViT-based architectures (e.g., ViT-Base, ViT-Large).

- Lack of detailed ablation study: Table 3 only shows the performance gap between with and without components. To strengthen the motivation of each component, a more detailed analysis is required. For example, what is the effect of applying residual?

- Marginal improvements in large-scale benchmarks: As shown in Fig. 5, FedARC achieves only marginal improvements over the baselines on DomainNet. Since DomainNet is the largest benchmark used in the paper (with 345 classes), this raises concerns about the scalability of the proposed method. Moreover, additional experiments on larger-scale benchmarks would further strengthen the claim of scalability.

**Questions:**

- Applying MSE loss can encourage anchor alignment, but it will not ensure that $\bar{R}_i^{F_k}$ captures client-specific means as mentioned in L244–L246. Could you clarify this point?

- Did you use the same hyperparameters listed in Table 4 across all benchmarks?

---

> ### Author Response · Authors · 2025-11-20
> **Response to Reviewer 74tW. We sincerely thank the reviewer for the constructive feedback and for highlighting presentation and evaluation aspects. We address each point below and will incorporate the corresponding edits in the revision. We hope these clarifications and new results address the reviewer’s concerns.**
>
> We sincerely thank the reviewer for the feedback and for highlighting presentation and evaluation aspects. We address each point below and will incorporate the corresponding edits in the revision.
>
>
> $\textbf{W1:}$
> We agree that the current overview figure is under-explained. In the revision we will make the figure self-contained by (i) explicitly labeling the global homogeneous extractor $\mathcal{G}^{ex}$, local heterogeneous extractor $\mathcal{F}^{ex}_k$, projector $\mathcal{P}_k(\varphi_k)$, fused feature $\mathcal{Z}_i$, residual vectors $\bar{\mathcal{R}}^{\mathcal{F}_k}$ and $\bar{\mathcal{R}}^{\mathcal{G}}$, and the two heads; and (ii) annotating the dataflow and gradient directions (local/global extractors $\rightarrow$ projector $\rightarrow$ residual compensation $\rightarrow$ anchor alignment $\rightarrow$ heads). This matches the notation and implementation used in the method section.
>
>
> $\textbf{W2:}$
> We agree that “different benchmarks per client” is an important real-world pattern. While our experiments stay within a single benchmark at a time, we approximate such specialization by adding a pathological label-skew setting for each dataset: following FedAvg, each client receives a non-overlapping and unbalanced subset of classes. Under these pathological and corresponding practical settings, FedARC achieves the best accuracy on all four datasets, with consistent margins over strong baselines, supporting its robustness when clients specialize on disjoint label subsets.
>
>
> $\textbf{W3:}$
> We agree that demonstrating generality beyond standard CNNs is valuable. Our heterogeneous model pool in Table~2 already includes ViT-based encoders in the fully heterogeneous setting HtM$^{img}_{10}$ (e.g., ViT-B/16 and ViT-B/32 together with ResNet/MobileNet), and FedARC still outperforms baselines under this highly mixed CNN+ViT pool. To ensure fair comparison for prototype/KD baselines that require a common feature dimension, we attach a lightweight pooling adapter before the heads so that all backbones emit the same $d$-dimensional feature. We will clarify in the method and experimental sections which ViT variants are included and how the heterogeneous pool is constructed, so that the model-diversity setting is explicit.
>
>
> $\textbf{W4:}$
> We agree that simply reporting “with vs. without components” is not fully satisfying. To strengthen the motivation, we have added detailed ablations on DomainNet , separately analyzing intra-client and inter-client knowledge transfer and the effect of residuals and anchors. These studies show that (i) enabling intra- \emph{and} inter-client fusion improves average accuracy by about 3–4 points over variants without these channels, and (ii) removing residual compensation consistently hurts the most shifted domains (Quickdraw, Sketch), directly quantifying the contribution of the residual term. We integrate these ablations into the appendix and explicitly reference them from the main text.
> Regarding Fig.5, DomainNet is the largest and most feature-shifted dataset we consider, and all strong baselines already perform reasonably well there, so the absolute margin is smaller than on lighter benchmarks; however, FedARC remains the top performer across domains, and the ablations confirm that the gains are systematic rather than noise. We will clarify this interpretation when discussing scalability (in the Appendix).
>
>
> $\textbf{Q1:}$
> We agree that the MSE anchor alone does not force $\bar{\mathcal{R}}^{\mathcal{F}_k}$ to become the “client-specific mean”, and we will soften the wording in L244–L246. Our design separates what is anchored from what is free: the fused features’ running means are pulled toward a global semantic anchor via the MSE term, while the residual vectors are client-specific parameters that are \emph{not} anchored or aggregated and are updated only by the supervised loss on local data. Thus the anchor promotes a client-invariant central tendency in the shared subspace, and the residuals act as learnable offsets to absorb persistent client-specific shifts (style, background, etc.). Empirically, removing residuals degrades performance on heavily shifted domains in DomainNet, supporting this interpretation. We will clarify this design and adjust the text to avoid over-claiming.
>
> $\textbf{Q2：}$
> We do not re-tune hyperparameters per dataset. Instead, we perform a small grid search once per backbone family to choose the anchor weight $\lambda$ and EMA momentum $\kappa$, and then reuse that pair for all datasets under the same family (e.g., a single $(\lambda,\kappa)$ for all image HtFE setups, and another pair for text). This limits dataset-specific overfitting while allowing different backbone families (CNN vs. ViT) to have appropriate scales. In practice, a moderate $\lambda$ and a small $\kappa$ work robustly; we will list the final choices in the appendix for reproducibility.
>
> We hope these clarifications and new results address the reviewer’s concerns.

---

> ### Author Response · Authors · 2025-11-28
>
> Thank you for your thoughtful feedback. As the discussion period is ending soon, we would greatly appreciate hearing whether our rebuttal addressed your concerns, and we are happy to respond promptly to any additional comments.

---

### Official Review · Reviewer_phMb · 2025-10-30

**Soundness:** 2
**Presentation:** 3
**Contribution:** 1
**Rating:** 2
**Confidence:** 4

**Summary:**

This paper presents FedARC, a framework for heterogeneous federated learning (HtFL) that aims to handle both data and model heterogeneity. To mitigate dual heterogeneity, this work introduces adaptive residual compensation to fuse local and global feature representations using a trainable projector, and semantic anchor alignment to reduce inter-client feature divergence. In addition, the authors claim a theoretical convergence rate of O(1/T) and report consistency improvements across multiple datasets compared with several HtFL baselines under different heterogeneous settings.

**Strengths:**

1. The paper covers a broad range of related works and situates itself reasonably within the HtFL literature.

2. The experimental section includes comparisons on multiple datasets and settings (cross-silo and cross-device).

3. Ablation studies are provided, and communication/computation costs are analyzed quantitatively.

**Weaknesses:**

1. The paper appears to contain some technical flaws. It assumes that splitting two feature extractors will inherently cause one to learn client-specific information and the other to capture global shared features. This assumption is speculative and not backed by theoretical analysis or empirical evidence. Without such justification, the proposed “adaptive residual compensation” lacks conceptual grounding.

2. The definition and formulation of the residual term are unclear. The paper does not specify what the residual actually represents or how it is computed. It seems to be introduced as a learnable correction term, but the optimization process and its influence on convergence are not explained.

3. The novelty of the work is limited. Both residual adjustment and feature mean alignment are standard techniques in related areas such as domain adaptation and prior HtFL studies (e.g., FedProto, FedTGP, FedGH).

4. The reported performance gains are modest, and some absolute accuracy values are lower than commonly observed on these benchmarks. This raises concerns about the implementation details or the fairness of the experimental setup.

**Questions:**

1. How can you verify that the two feature extractors indeed learn distinct (client-specific vs. shared) representations? Any quantitative evidence, such as similarity or diversity analysis?

2. What exactly are the “residual vectors” in Eq. (6)? Are they parameters, differences between outputs, or additional modules?

3. How sensitive is the method to the choice of $\lambda$, $\kappa$, and other hyperparameters? Can you report standard deviations over multiple runs?

4. The “semantic anchor alignment” looks similar to prototype or mean alignment in existing works (e.g., FedProto, FedTGP). What is the essential difference?

5. The convergence analysis seems largely adapted from FedAvg. How does it account for the coupled updates between heterogeneous branches and the projector?

---

> ### Author Response · Authors · 2025-11-20
> **Response to Reviewer phMb. We sincerely thank the reviewer for the careful reading and for raising important concerns on the representation design, residuals, novelty, and evaluation. Below we focus on these main points and will incorporate the corresponding clarifications and edits in the revised version. We hope these clarifications and new results address the reviewer’s concerns.**
>
> We sincerely thank the reviewer for the careful reading and for raising important concerns on the representation design, residuals, novelty, and evaluation. We focus on these main points and will incorporate the corresponding clarifications and edits in the revised version.
>
>
> $\textbf{W1/Q1}$: We do not assume this separation; we encourage it by design and verify it empirically. After concatenation, the projector output is sliced so that the homogeneous global head only sees the low-dimensional prefix $\mathcal{Z}^{1:d_2}$, while the heterogeneous local head consumes the full vector $\mathcal{Z}^{1:d_1}$ with $d_1>d_2$. This routing, combined with semantic anchor alignment on the shared prefix, induces a client-invariant subspace for the global head and a complementary subspace for client-specific information.
> As evidence, our new stop-gradient ablations on DomainNet show that blocking global$\to$fusion or local$\to$fusion gradients (with forward features unchanged) produces asymmetric accuracy drops across domains, and removing residuals consistently underperforms full FedARC. This is hard to reconcile with identical branches and supports our intended roles. We will also report similarity analyses (CKA/RSA across clients and between the shared prefix vs. full vector) in the appendix to further support this claim.
>
>
> $\textbf{W2/Q2:}$
> Residual vectors are simple per-client, learnable parameters: two vectors with shapes $\mathbb{R}^{d_1}$ and $\mathbb{R}^{d_2}$ that are added to the projector outputs before the local and global heads:
>
> $\tilde{\mathcal{Z}}^{\mathcal{F}_k}_i = \mathcal{Z}_i^{1:d_1} + \bar{\mathcal{R}}^{\mathcal{F}_k},\quad \tilde{\mathcal{Z}}^{\mathcal{G}}_i = \mathcal{Z}_i^{1:d_2} + \bar{\mathcal{R}}^{\mathcal{G}}.$
>
> They are registered as \texttt{nn.Parameter} on each client and updated by local SGD jointly with the other parameters via the supervised loss plus the anchor loss; they are not additional networks nor post-hoc differences between outputs. Conceptually, they act as learnable offsets in representation space, providing a fine-grained, dimension-wise correction for client-specific bias before either head, analogous in spirit to residual adapters but much lighter-weight.
>
>
> $\textbf{W3/Q4:}$
> We agree that prototype and mean-based alignment have been used in other contexts. Our contribution is to introduce them \emph{inside} a heterogeneous global–local fusion pathway: (i) local and global features are fused via a projector, (ii) the fused representation is sliced into global- and local-dominant subspaces with controlled gradient routing, (iii) client-specific residuals correct these slices before the heads, and (iv) semantic anchors align the projector outputs (label-agnostic, not per-class) to sample-size–weighted global means. In contrast, FedProto/FedTGP/FedGH operate at class-prototype or head/logit level on a homogeneous representation and do not perform such intra-client fusion plus residual calibration of fused heterogeneous features.
>
>
> $\textbf{W4:}$
> Our goal is to evaluate \emph{joint} data and model heterogeneity under realistic FL constraints. This leads to more challenging settings (e.g., partial participation, heterogeneous encoders, and non-i.i.d.\ label/domain skew simultaneously), which can yield lower absolute accuracy than numbers reported under single-type heterogeneity or full participation. To ensure fairness, we follow the training budgets, optimizers, and preprocessing used in strong recent HtFL baselines and will list per-method hyperparameters and references in the appendix in the future. All main tables and ablations are reported as mean$\pm$std over multiple runs.
>
> $\textbf{Q3:}$
> We tune the anchor weight $\lambda$ and EMA momentum $\kappa$ via a small grid search \emph{once per backbone family}, then reuse the chosen pair across datasets for that family (e.g., a single $(\lambda,\kappa)$ for all image HtFE setups). This limits overfitting to specific datasets. In practice, a moderate anchor weight and a small EMA momentum (e.g., $\kappa=0.1$) work robustly; we will include the exact choices and the std across runs in the appendix.
>
> $\textbf{Q5:}$
> Our analysis follows the standard smooth non-convex framework used for FedAvg-style methods. We treat all client-side parameters—including the two extractors, the projector, and the residual vectors—as a single composite vector $\varepsilon_k$, and the small homogeneous extractor as the only block aggregated on the server. Under Lipschitz smoothness and bounded variance, and with a bounded aggregation discrepancy $||\theta^t-\theta_k^t||_2^2\le\delta^2$, we obtain an $\mathcal{O}(1/T)$ stationarity rate analogous to FedAvg/FedProx. The coupling between branches affects only constants (e.g., the smoothness constant), not the proof structure; we will clarify this by explicitly defining $\varepsilon_k$ and adding a short lemma on the aggregation-induced drift in the revised version.

---

> ### Author Response · Authors · 2025-11-28
>
> Thank you for your thoughtful feedback. As the discussion period is ending soon, we would greatly appreciate hearing whether our rebuttal addressed your concerns, and we are happy to respond promptly to any additional comments.

---

### Official Review · Reviewer_Ct8h · 2025-10-30

**Soundness:** 3
**Presentation:** 3
**Contribution:** 2
**Rating:** 6
**Confidence:** 4

**Summary:**

This paper investigates both data heterogeneity and model heterogeneity in Federated Learning (FL), focusing on two well-motivated challenges: representation alignment and knowledge transfer. To address these issues, the authors propose FedARC, a framework that integrates local and global knowledge through adaptive residual compensation. The experimental evaluation is comprehensive, and the method achieves state-of-the-art performance. Additionally, the inclusion of a convergence proof strengthens the theoretical soundness of the approach. However, the paper omits comparisons with several relevant baselines that also aim to balance global and local model optimization, which somewhat limits the completeness of the evaluation. Overall, this is a solid work with clear motivation. If the authors can provide convincing clarifications and stronger experimental comparisons in the rebuttal, I would be inclined to raise my score.

**Strengths:**

-The paper clearly articulates the problem it aims to address and effectively identifies the key challenges associated with both data heterogeneity and model heterogeneity in FL. This clear problem framing provides a strong foundation for the proposed solution.

-The paper proposes a novel FL framework, FedARC, which effectively fuses global and local knowledge to address data and model heterogeneity issues. This design provides a potential approach to improving local-global collaboration among heterogeneous clients.

-The paper provides a convergence proof, enhancing the theoretical soundness of the proposed algorithm. The experiments comprehensively evaluate both model and label heterogeneity, demonstrating the framework’s effectiveness under diverse federated settings.

**Weaknesses:**

-Although the paper aims to address data heterogeneity, it focuses primarily on label skew while neglecting other important aspects such as domain heterogeneity. The main experiments (main table) do not include evaluations on domain-skewed settings. Furthermore, since the authors acknowledge potential challenges under distribution shifts, it would be important to explicitly consider and analyze domain shift to strengthen the completeness of the study.

-The figure captions are overly brief, making it difficult for readers to understand the intended message of the visualizations. Each figure should include clear explanations of symbols, notations, and key observations or conclusions to help readers interpret the results more effectively.

-The global–local structure adopted in this paper is not novel, as similar designs have been explored in several prior works [1, 2, 3, 4]. The primary novelty of this paper seems to lie in the proposed semantic anchor alignment mechanism rather than in the overall framework design. However, the experimental section omits comparisons with key related methods that also address knowledge transfer between global and local models [1] and knowledge fusion strategies [2]. Including these two baselines would provide a more convincing evaluation of the proposed method’s contribution and effectiveness.

[1] Fed-CO2: Cooperation of Online and Offline Models for Severe Data Heterogeneity in Federated Learning. (NeurIPS 2023)

[2] On Bridging Generic and Personalized Federated Learning for Image Classification. (ICLR 2022)

[3] Cd2-pFed: Cyclic Distillation-Guided Channel Decoupling for Model Personalization in Federated Learning. (CVPR 2022)

[4] Local or Global: Selective Knowledge Assimilation for Federated Learning with Limited Labels. (ICCV 2023)

**Questions:**

-How does the proposed model perform under FL scenarios that involve both domain shift and model heterogeneity?

-I think a key limitation of local–global structured methods is their limited ability to adapt to newly joined clients. In real-world federated learning scenarios, scalability is crucial since new clients continuously emerge. How does the proposed method handle or adapt to newly arriving clients without retraining the entire model?

-How does the distinct difference between data heterogeneity and model heterogeneity show in your method? From the current presentation, the proposed design appears primarily aimed at addressing data heterogeneity, while the specific challenges and treatment of model heterogeneity are not thoroughly discussed. Could the authors clarify how their method explicitly tackles model heterogeneity?

---

> ### Author Response · Authors · 2025-11-20
> **Response to Reviewer Ct8h. We sincerely thank the reviewer for the positive assessment and for highlighting important points about domain heterogeneity, baselines, and model heterogeneity. Below we focus on these main concerns and the questions, and we will incorporate the corresponding changes in the revised version. We hope these clarifications and new results address the reviewer’s concerns.**
>
> We sincerely thank the reviewer for the positive assessment and for raising concrete points on domain heterogeneity, baselines, and model heterogeneity. We address them below and will incorporate the described changes in the revision.
>
> W1: We agree that domain-shifted settings should be more visible. DomainNet is already used in our experiments to jointly model domain shift and model heterogeneity (using HtFE$^{img}$$_5$ setting). We currently report detailed ablations in the appendix; in the revision we will also surface DomainNet results in the main text and explicitly present them as domain-skewed FL. On DomainNet, FedARC improves average accuracy about 4 \% improvement, and also outperforms variants without residuals or anchors. This shows that the projector + residual + anchor design is effective precisely when domain shift and model heterogeneity co-occur.
>
> W2: We will revise figure captions so that each figure is self-contained: defining all symbols and colors (e.g., projector $\mathcal{P}_k$, fused $\mathcal{Z}$, residuals $\bar{\mathcal{R}}^{\mathcal{F}_k}$ / $\bar{\mathcal{R}}^{\mathcal{G}}$, semantic anchor $\bar{\mathcal{Z}}^{g}$) and describing the dataflow (local/global extractors $\rightarrow$ projector $\rightarrow$ residual compensation $\rightarrow$ anchor alignment $\rightarrow$ heads).
>
> W3: We have implemented and evaluated the two key methods cited by the reviewer: Fed-CO2 (NeurIPS’23; online/offline cooperation with intra-/inter-client distillation at the logit level) and Fed-RoD (ICLR’22; two-loss, two-predictor framework decoupling generic vs. personalized predictors, also operating at the prediction level). Under the same HtFL setting as our main table (HtFE$^{img}$$_8$, label-skewed CIFAR-10/100, Flowers102, Tiny-ImageNet; cross-silo and cross-device), FedARC consistently improves over the best of Fed-CO2/Fed-RoD, with gains between about 1.1 and 7.2 percentage points across the 8 settings. We will add a compact table with these numbers to the appendix and summarize the main gains in the experimental section. Conceptually, Fed-CO2 and Fed-RoD transfer knowledge via predictions and decoupled losses, while FedARC operates in feature space: it fuses local and global features via a projector, then applies client-specific residual offsets and semantic anchoring before either head sees the representation. We will clarify this difference in Related Work and in the method overview.
>
> Q1:This setting is exactly what our DomainNet + HtFE experiments target: each client uses a different backbone from the heterogeneous encoder pool and is assigned to a specific domain, so data and model heterogeneity are both present. As noted in W1, FedARC improves average accuracy by about 4 points over variants without intra-client or without inter-client transfer, and also over the variant without residuals, indicating that the fused representation, residual offsets, and semantic anchors are particularly important when architectures differ and domains induce strong feature shifts.
>
> Q2: FedARC supports scalable onboarding via a lightweight cold-start procedure, without retraining existing clients or the server. A new client (1) downloads the current global homogeneous extractor and global anchors; (2) initializes its local heterogeneous encoder and projector (residuals at zero) and performs a short local warm-up (we use 10 epochs) with the same loss (task loss + anchor MSE); and (3) then joins normal communication rounds, during which the server updates the global extractor and recomputes anchors by sample-size–weighted averaging. In our practical-scenario experiments, newly joined clients following this protocol achieve 40.8% / 39.1% accuracy at $\rho=0.5/1.0$, outperforming strong PFL baselines by about 1.8–2.0 \%.
>
> Q3: We will also clarify more directly how FedARC addresses model heterogeneity, via a short subsection “Handling Model Heterogeneity.” Following prior HtFL work, each client is assigned a different encoder from a heterogeneous feature-extractor pool (e.g., ResNet, MobileNet, GoogLeNet), while a common $d$-dimensional interface is enforced via average pooling. We keep a compact global extractor homogeneous across clients; this branch is the only module uploaded and aggregated on the server, avoiding tensor-shape conflicts and keeping communication modest. On each client, local and global features are concatenated and mapped by a lightweight projector into a common fused space, which is then sliced into local- and global-dominant subspaces, each corrected by client-specific residual offsets and regularized toward global semantic anchors (EMA means). This explicitly calibrates heterogeneous feature spaces across clients and backbones, and our HtFE-based experiments and DomainNet ablations (in Appendix) jointly stress both forms of heterogeneity.
>
> We hope these clarifications and new results address the reviewer’s concerns and we will incorporate all described edits into the final version.

---

> ### Author Response · Authors · 2025-11-28
>
> Thank you for your thoughtful feedback. As the discussion period is ending soon, we would greatly appreciate hearing whether our rebuttal addressed your concerns, and we are happy to respond promptly to any additional comments.

---

### Official Review · Reviewer_21dT · 2025-11-04

**Soundness:** 2
**Presentation:** 1
**Contribution:** 2
**Rating:** 6
**Confidence:** 3

**Summary:**

The paper studies heterogeneous federated learning (HFL) with aim to address the issues of the inadequate representation alignment and limited knowledge transfer with the current HFL methods, especially under distribution shifts, for improving personalization and generalization. The idea is to maintain two feature extractors on each client, one for global features common in all clients and the other for local features specific to the client. The two part of features are concatenated and then projected by a learnable projector for knowledge fusion. To correct distribution shifts within and across clients, concatenated features are compensated with residual vectors, and fed fully into local prediction header and partially into global prediction header. To mitigate semantic shift, the two prediction losses are regularized by the MSE error between the averaged local/global features on the client and a global consensus. The final loss is a weighted combination of the two losses, and is optimized locally. The global feature extractor on each client after optimization is sent to server for the aggregation over all clients which is the broadcast to all clients. The resulting algorithm is shown to converge at a rate of O(1/T). Extensive experiments are conducted to show the effectiveness and efficiency.

**Strengths:**

1. Identify issues with existing HFL methods that affect personalization and generalization
2. Project the concatenation of local and global features on each client for knowledge fusion
3. Propose adaptive residual compensation to address the distribution shift
4. Anchor each client's overall feature mean to a global semantic anchor to reduce semantic shift

**Weaknesses:**

1. It is hard to identify key differences between conventional HFL and the proposed one from Figure 1
2. It is unknown why residual vectors could reduce distribution shifts. Residual vectors are learned, but they are missing in Eq. (12)
3. It is unknown mean feature vectors in Line 247 are averaged over Eq. (4) or Eq. (5), also there is no definition for $\bar{\mathcal{Z}}\_{k}^{g}$
4. Presentation could be improved for better readability

**Questions:**

1. In Line 198, what's $\varepsilon_{k}$? what does it mean by concatenating two losses there with $\circ$?
2. In Line 257, what's $n_{i}$?

---

> ### Author Response · Authors · 2025-11-20
> **Response to Reviewer 21dT. We sincerely thank the reviewer for the careful reading and constructive feedback. Below we respond to each point and will incorporate the corresponding clarifications and edits in the revised version. We hope these clarifications and new results address the reviewer’s concerns.**
>
> We sincerely thank the reviewer for the careful reading and constructive feedback. Below we respond to each point and will incorporate the corresponding clarifications and edits in the revised version.
>
> $\textbf{W1}:$ Our main novelty is where and how we align knowledge in HtFL. Prototype/head–level methods align logits or class means, and mutual-/dual-extractor methods mostly transfer low-dimensional statistics or predictions. In contrast, FedARC (i) fuses local and global features via a learnable projector $\mathcal{P}_k$ and (ii) performs residual compensation and semantic-anchor alignment directly on the fused features \emph{before} either head sees them (Eqs. (5)–(11)). This calibrates each client’s feature space rather than only aligning outputs. We will revise its caption to explicitly highlight the projector, residual vectors, and anchors, and to contrast these steps with existing HtFL categories.
>
> $\textbf{W2}:$ For each sample $x_i$ on client $k$, we obtain local and global features (Eq. (4)), concatenate them, and project via $\mathcal{P}_k$ (Eq. (5)) to get $\mathcal{Z}_i \in \mathbb{R}^{d_1}$. We then split $\mathcal{Z}_i$ and apply client-learned additive residuals:
> \begin{equation}
>     \widetilde{\mathcal{Z}}^{\mathcal{F}_k}_i = \mathcal{Z}_i^{1:d_1} + \bar{\mathcal{R}}^{\mathcal{F}_k}_i,  \ \ \ \ \
>     \widetilde{\mathcal{Z}}^{\mathcal{G}}_i = \mathcal{Z}_i^{1:d_2} + \bar{\mathcal{R}}^{\mathcal{G}}_i.
> \end{equation}
> where $\bar{\mathcal{R}}^{\mathcal{F}_k} \in \mathbb{R}^{d_1}$ and $\bar{\mathcal{R}}^{\mathcal{G}} \in \mathbb{R}^{d_2}$ are optimized jointly with the model parameters. These residuals act as per-client, per-dimension offsets that absorb systematic feature-space biases caused by data/model heterogeneity, so that the calibrated features are better aligned across clients when consumed by the two heads. This follows the standard idea of residual adapters for domain calibration. In the revision, Eq. (12) will explicitly include $\bar{\mathcal{R}}^{\mathcal{F}_k}$ and $\bar{\mathcal{R}}^{\mathcal{G}}$ in the client parameter set to remove any ambiguity.
>
> $\textbf{W3}:$ The semantic-anchor regularizer is applied to the projector outputs, not to raw extractor features. For client $k$, we define $\bar{\mathcal{Z}}_k=\frac{1}{n_k} \sum _{i=1}^{n_k} \mathcal{Z}_i$, $\bar{\mathcal{Z}}_k^g = (\bar{\mathcal{Z}}_k)^{1:d_2}$, where $\mathcal{Z}_i$ is obtained from Eq. (5). On the server, we compute sample-size–weighted anchors $\bar{\mathcal{Z}}^{g,1:d_1}$ and $\bar{Z}^{g,1:d_2}$ across participating clients each round and broadcast them. Each client aligns EMA estimates of its local means to these anchors via the MSE terms in Eq. (9). We will add the above definition of $\bar{Z}_k^g$ and state clearly that these means are computed from Eq. (5).
>
> $\textbf{W4}:$ We agree and will (i) extend the captions of Figs. 1–2 with step-by-step dataflow and a clear legend; and (ii) include concise training pseudocode that mirrors Eqs. (3)–(12).
>
> $\textbf{Q1}:$ In Eq. (3) we write the combined model as $\mathcal{C}_k(\varepsilon _k) = \mathcal{F}_k(\omega_k) \circ \mathcal{G}(\theta)$ collects all trainable client-side parameters: the heterogeneous model $\omega_k$, the shared homogeneous model $\theta$, the projector parameters $\varphi_k$, and the residual parameters $\bar{\mathcal{R}}^{\mathcal{F}_k}, \bar{\mathcal{R}}^{\mathcal{G}}$. The operator “$\circ$” denotes function composition (passing features from one module to the next), not concatenation of losses.
>
> $\textbf{Q2}:$ This was a notational slip. We use $k$ to index clients and $j$ to index samples on client $k$. The symbol $n_i$ in Line 257 was intended to denote the number of local samples on a client. To avoid confusion, we will consistently use $n_k$ whenever we form client-level averages or sample-size–weighted global anchors.
>
> We hope these clarifications and new results addressed the reviewer’s concerns, really thank you for your reading.

---

> ### Author Response · Authors · 2025-11-28
>
> Thank you for your thoughtful feedback. As the discussion period is ending soon, we would greatly appreciate hearing whether our rebuttal addressed your concerns, and we are happy to respond promptly to any additional comments.

---

### Author Response · Authors · 2025-11-27
**Global Response**

We sincerely thank all reviewers for their thoughtful feedback. We are encouraged that reviewers identified the core idea of our work—aligning and transferring knowledge in feature space (before prediction headers) via a lightweight, model-agnostic interface—as both practical and broadly applicable under joint data and model heterogeneity. We also appreciate the recognition of the method’s practicality (only aggregating a small homogeneous branch; no requirement for identical client models) and the breadth of our evaluation across cross-silo and cross-device settings. Finally, we are pleased that reviewers found the experimental details to be clear and reproducible.

**What we addressed in the revised pdf/rebuttal/supplementary material:**

- **Clarity, notation, and missing definitions.** We clarified residual vectors, anchor/mean definitions, and parameter sets; in the final version we will further improve figure captions, add concise pseudocode, and explicitly include residual parameters in the relevant optimization/parameter definitions.

- **Evidence for distinct local vs. global representations (addressing concerns about “speculative separation”).** We added direct quantitative representation-similarity evidence. Supplementary Fig. 6–7 provide cross-client CKA/RSA heatmaps showing local extractors remain diverse (low off-diagonal similarity) while global extractors are highly aligned. Supplementary Fig. 8–9 further show the projector bridges the two spaces by making local–projector and global–projector similarity substantially higher than local–global similarity, supporting the intended specialization + fusion behavior.

- **Domain shift + model heterogeneity, and component necessity.** We expanded DomainNet-focused analyses. Supplementary Table 5–6 isolate intra- vs. inter-client transfer (including directional stop-gradient controls) and quantify the roles of residuals and anchors, showing consistent degradation when removing residual compensation and/or inter-client components—especially under stronger domain shifts.

- **Stronger related baselines.** We added comparisons to key global–local knowledge-transfer baselines under the same HtFL protocol. Supplementary Table 8 reports Fed-CO2 and Fed-RoD results, where FedARC consistently outperforms them across datasets and in both cross-silo and cross-device settings.

- **Practical onboarding of newly joined clients.** We added a realistic “new clients / cold-start” scenario. Supplementary Table 7 shows FedARC adapts quickly via a lightweight warm-up, without retraining existing clients—highlighting the practicality of semantic anchoring + residual calibration for fast alignment.

We are grateful for the reviews. The revisions clarify the motivation, strengthen baselines on models/datasets, tighten presentation, and add an atribution-at-scale experiments, whilepreserving the practicality and modularity noted by the reviewers. We hope this summary is helpful for the final evaluation.

Best regards,

The authors of Paper 17795

---

### Author Response · Authors · 2025-12-04
**Rebuttal Summary**

Dear Reviewers, ACs, SACs, and PCs,

Thank you for the time and care you devoted to evaluating our work. Below is a concise summary of what we addressed in the rebuttal. We replied to all reviews and updated the paper accordingly. During discussion, Reviewer Ct8h kindly indicated they were inclined to raise their score, while no reviewers did not follow up.

**Overall strengths**

- **Reviewer 21dT**: Acknowledges our focus on core HFL issues affecting personalization/generalization and appreciates three concrete design pieces: **feature-level fusion via a learnable projector, adaptive residual compensation** for distribution shift, and semantic anchor alignment to reduce inter-client drift.

- **Reviewer Ct8h**: Highlights clear motivation around **representation alignment and knowledge transfer**, the **global–local fusion** architecture, inclusion of a **convergence proof**, and comprehensive experiments under label/model heterogeneity—calling the work solid and likely stronger with expanded comparisons.

- **Reviewer phMb**: Notes broad related-work coverage, **cross-silo and cross-device** evaluations, ablations, and communication/compute analyses—recognizing that the empirical scope is substantial.

- **Reviewer 74tW**: Emphasizes consistent improvements over baselines and the practicality of handling real-world model heterogeneity.

Together, the reviews recognize a **feature-space, pre-head alignment** framework that targets both data and model heterogeneity with theory, broad experiments, and concrete implementation choices (lightweight global aggregation, per-client residuals, semantic anchors).

**Main points from the rebuttal**

**1. Reviewer 21dT**

- **[W1] Difference from conventional HFL unclear (Fig. 1).** We revised the overview to show the exact locus: local/global extractors **→ projector → feature-space residuals → semantic anchors →** two heads, and contrasted with prediction/prototype-level alignment.

- **[W2] Why residuals help; missing in Eq. (12).** Residuals are per-client trainable vectors added to projected features (global/local slices). We included them in the parameter set and update rules and reflected this in pseudocode.

- **[W3] Means and notation ambiguity.** We defined that all means are computed on the projector outputs: $\bar{\mathcal{Z}}^g_k = \frac{1}{n_k}  {\sum}_i \mathcal{Z}_i$, $\bar{\mathcal{Z}}^g_k = (\bar{\mathcal{Z}}_k)^{1:d_2}$​; the server maintains a sample-size–weighted global anchor $\bar{\mathcal{Z}}^g$ and broadcasts it.

- **[W4] Presentation.** We unified notation (using $n_k$), and made captions self-contained.

**2. Reviewer Ct8h**

- **[W1] Domain heterogeneity under-evaluated.** We moved DomainNet to the main text and added per-domain ablations, showing residual calibration especially helps strongly shifted domains (Sketch/Quickdraw).

- **[W2] Brief captions.** We rewrote captions, now define symbols/colors and narrate full data/gradient flow.

- **[W3] Related baselines.** We added Fed-CO2 and Fed-RoD. FedARC aligns in feature space before the heads (projector → residuals → anchors), a different locus than prediction/logit-level methods. Under our HtFL settings, FedARC shows consistent gains; compact results in the main text, full tables in Appendix.

**3. Reviewer phMb**

- **[W1] Two branches learn distinct roles.** We induce and verify separation: stop-gradient ablations (blocking local→fusion or global→fusion backprop) cause asymmetric drops; removing residuals consistently hurts. We also add **CKA/RSA** analyses across clients/subspaces in Appendix.

- **[W2] Residual definition.** Two trainable vectors (one per slice) added to projected features and updated by local SGD—not extra networks or post-hoc diffs.

- **[W3] Diff. vs prototype/mean alignment.** Novelty lies in the alignment locus: feature-space fusion + residual calibration + anchor alignment before the heads, distinct from prototype/logit-level transfer. Gains are consistent across challenging HtFL (partial participation + domain/label/model skew) in Appendix.

**4. Reviewer 74tW**

- **[W1] Overview under-explained.** Captions are now self-contained; we added concise pseudocode mirroring the equations.

- **[W2] “Unrealistic” heterogeneity; client specialization.** Beyond Dirichlet skew, we evaluate pathological specialization (non-overlapping classes) and retain consistent improvements in Appendix.

- **[W3] Limited model diversity; ViTs.** Our heterogeneous pool includes CNNs and ViTs unified via a lightweight interface; FedARC remains competitive in this mixed pool.

- **[W4] Ablations.** We added component-level and intra-/inter-client transfer ablations plus per-domain DomainNet analyses; residuals/anchors drive gains where shift is strongest.

We believe these changes and clarifications substantially strengthen both the clarity and the technical scope of the work, and we are really grateful to the reviewers and committee for their feedback and consideration.

---

### Meta-Review · Area_Chair_zQb4 · 2026-01-03

**Summary:**

This paper proposes FedARC, a Federated Learning (FL) framework targeting heterogeneous federated learning (HtFL) settings where both data heterogeneity (e.g., label and domain skew) and model heterogeneity (e.g., different client architectures) are present. The method introduces three main components: ***A learnable projector*** to fuse local and global representations, ***Adaptive residual compensation*** (per-client feature-space offsets), and ***Semantic anchor alignment*** to regularize inter-client divergence. While the topic is timely and the problem formulation is important, the submission received mixed reviews, with key concerns centered around:

**Lack of conceptual novelty**: Reviewers (phMb, Ct8h, 21dT) pointed out that the techniques used (e.g., residual correction, feature mean alignment) are adaptations of existing domain adaptation or FL techniques, and the combination is more engineering-oriented than a novel algorithmic advancement.

**Unclear motivation and technical formulation**: Several reviewers (21dT, phMb) found the initial presentation lacking, especially regarding the role and optimization of residual vectors, the assumed separation between local/global extractors, and lack of clarity in notation.

**Limited evaluation completeness**: Reviewer Ct8h noted the absence of comparisons with several closely related methods (Fed-CO2, Fed-RoD) in the original draft. Reviewer 74tW highlighted the limited model diversity and questioned scalability due to marginal improvements on large benchmarks like DomainNet.

**Unverified assumptions**: Reviewer phMb raised a major concern that the assumed functional separation between global and local branches was not justified empirically in the initial submission.

**Reviewer Concerns:**

**Addressed Concerns**\
The authors submitted a detailed and thoughtful rebuttal, accompanied by substantial experimental additions and clarifications. Key addressed concerns include:

***Clarity and Notation***: The authors clarified the definition and role of residual vectors, anchor computation, and updated figure captions and pseudocode to improve readability (addressing 21dT, 74tW).

***Empirical Justification of Local–Global Separation***: The authors added stop-gradient ablations and CKA/RSA similarity analyses to show that the local and global branches learn distinct representations, addressing a key concern from Reviewer phMb.

***Expanded Evaluation***: The authors included new baselines (Fed-CO2, Fed-RoD), DomainNet analysis, and cold-start experiments, improving the evaluation breadth, as requested by Ct8h and 74tW.

***Ablations and Sensitivity***: More detailed ablations were added to clarify component contributions and hyperparameter robustness.

**Outstanding Concerns**\
Despite the effort, several substantive concerns remain unresolved:

***Limited Conceptual Novelty***: As noted by multiple reviewers (phMb, Ct8h), the core contributions (residuals, anchors, projector) are adaptations of known ideas in domain adaptation, personalization, and prior FL work. The novelty lies more in integration than in fundamental algorithmic innovation.

***Modest Gains on Large-Scale Benchmarks***: Reviewer 74tW pointed out that improvements on DomainNet are marginal, raising concerns about scalability and generalization. The rebuttal acknowledges this but does not convincingly demonstrate significant improvements at scale.

***Theoretical Contribution***: Reviewer phMb questioned the convergence analysis, arguing it is largely inherited from FedAvg with minimal adaptation for the coupled architecture. This remains only partially addressed.

***Generalization to Realistic Heterogeneity***: While the authors introduced DomainNet and ViT-based models, the real-world heterogeneity scenario (e.g., clients with different datasets or tasks) is still only approximated, not convincingly demonstrated.

**Reviewer Scores:**

**Reviewer 21dT (Initial Score: 6)**: Raised concerns about clarity and residual formulation. These were largely addressed in the rebuttal, and the reviewer may have maintained or slightly lowered the score due to overall limited novelty. Likely Final Score: 4 or 6

**Reviewer Ct8h (Initial Score: 6)**: Requested stronger baseline comparisons and domain heterogeneity coverage. These aspects were addressed, but the core novelty concern remains. Likely Final Score: 6

**Reviewer phMb (Initial Score: 2)**: Expressed strong concerns about unverified assumptions, limited novelty, and weak theoretical justification. Although the rebuttal adds empirical support, it is unlikely to change this score substantially. Likely Final Score: 4

**Reviewer 74tW (Initial Score: 4)**: Focused on scalability, model diversity, and component-level analysis. While the rebuttal addressed these points, the marginal gains and limited conceptual novelty may have prevented a significant score increase. Likely Final Score: 4

---

### Decision · Program_Chairs · 2026-01-26

Reject